# Negative dielectric constant of water confined in nanosheets

Akira Sugahara [1], Yasunobu Ando[2,3], Satoshi Kajiyama [1], Koji Yazawa[4], Kazuma Gotoh[3,5], Minoru Otani[2,3], Masashi Okubo[1,3] & Atsuo Yamada [1,3]

Electric double-layer capacitors are efficient energy storage devices that have the potential to account for uneven power demand in sustainable energy systems. Earlier attempts to improve an unsatisfactory capacitance of electric double-layer capacitors have focused on meso- or nanostructuring to increase the accessible surface area and minimize the distance between the adsorbed ions and the electrode. However, the dielectric constant of the electrolyte solvent embedded between adsorbed ions and the electrode surface, which also governs the capacitance, has not been previously exploited to manipulate the capacitance. Here we show that the capacitance of electric double-layer capacitor electrodes can be enlarged when the water molecules are strongly confined into the two-dimensional slits of titanium carbide MXene nanosheets. Using electrochemical methods and theoretical modeling, we find that dipolar polarization of strongly confined water resonantly overscreens an external electric field and enhances capacitance with a characteristically negative dielectric constant of a water molecule.

[1] Department of Chemical System Engineering, School of Engineering, The University of Tokyo, Hongo 7-3-1, Bunkyo-ku, Tokyo 113-8656, Japan. [2] CD-FMat, National Institute of Advanced Industrial Science and Technology (AIST), Umezono 1-1-1, Tsukuba, Ibaraki 305-8568, Japan. [3] Elements Strategy Initiative for Catalysts and Batteries (ESICB), Kyoto University, Nishikyo-ku, Kyoto 615-8245, Japan. [4] JEOL Resonance, 3-1-2 Musashino, Akishima, Tokyo 196-8558, Japan. [5] Graduate School of Natural Science and Technology, Okayama University, 3-1-1 Tsushima-naka, Okayama 700-8530, Japan. These authors contributed equally: Akira Sugahara, Yasunobu Ando. Correspondence and requests for materials should be addressed to A.Y. (email: yamada@chemsys.t.u-tokyo.ac.jp)

An electric double-layer capacitor (EDLC) is an important class of electrochemical capacitors, in which electrochemical double layers are formed on the electrode surface and polarized solvents between ions and the electrode act as a dielectric medium[1,2]. Due to the fact that the electrochemical double-layer formation is a fast and highly reversible surface process with minimal ion migration, EDLCs can operate intrinsically at high charge/discharge rates without degradation over millions of repeated cycles, which enables load-leveling of intermittent power from renewable energy sources. However, as capacitance of currently used EDLC electrodes is limited, energy density of EDLCs is not satisfactory to be widely distributed in power grids. Therefore, increasing the capacitance of EDLC electrodes has been an active area of research[3–8].

From a theoretical point of view, a conventional EDLC electrode can be considered as a parallel-plate capacitor that delivers a capacitance ($C$) according to

$$C = \frac{\varepsilon A}{d} \qquad (1)$$

where $\varepsilon$ is the permittivity between the ions and the electrode, $A$ is the electrode surface area, and $d$ is the separation between the ions and the electrode (Supplementary Table 1)[1]. Consequently, increasing $A$ of the EDLC electrodes using meso- or nanostructured materials is a classic approach toward realizing large specific capacitance and high energy density[3–7]. Alternatively, in 2006, Chmiola et al.[8] experimentally proved the confinement effect on $d$ in microporous carbon EDLCs (pore size < 1 nm) that achieves an anomalous increase in the specific capacitance from 95 to 140 F g$^{-1}$[8–10].

Two-dimensional materials, such as graphene sheets[11], transition metal dichalcogenides[12], and MXenes[5], have recently been developed as electrode materials in EDLCs. Complete delamination of the two-dimensional materials increases the accessible surface area and the nanosheets with a large interlayer separation give a large specific capacitance greater than 240 F g$^{-1}$[5]. However, except for a few theoretical simulation results[13], there has been very limited research carried out on the application of the confinement effect in two-dimensional materials as an additional strategy toward high-energy density capacitors.

Our aim was to quantify the contribution of the confinement effect to capacitance in EDLC electrodes consisting of two-dimensional materials, theoretically modeled as a slit capacitor (Fig. 1a)[9,13]. This conceptually straightforward but experimentally difficult study was performed using MXene EDLC electrodes. As pioneered by research groups led by Gogotsi and colleagues[5,14,15], MXene is a class of two-dimensional materials with the following chemical formula: $M_{n+1}X_nT_x$ (where M is a transition metal, X is carbon or nitrogen, T is surface termination groups) and gives a large capacitance that is associated with ion intercalation[16]. Importantly, MXene maintains its stacked structure with a short interlayer distance during ion (de)intercalation, owing to strong interactions among the surface termination groups, intercalated ion species, and embedded solvents[17]. Such an anomalous interlayer interaction has led us to expect that MXene nanosheets strongly confine the intercalated ions and MXene is an ideal platform to study the guest confinement effect in two-dimensional materials.

Herein, using the MXene EDLC electrode as experimental and theoretical models for a slit capacitor, we demonstrate the negative dielectric constant of water confined between MXene nanosheets. This specific dipolar polarization of the confined

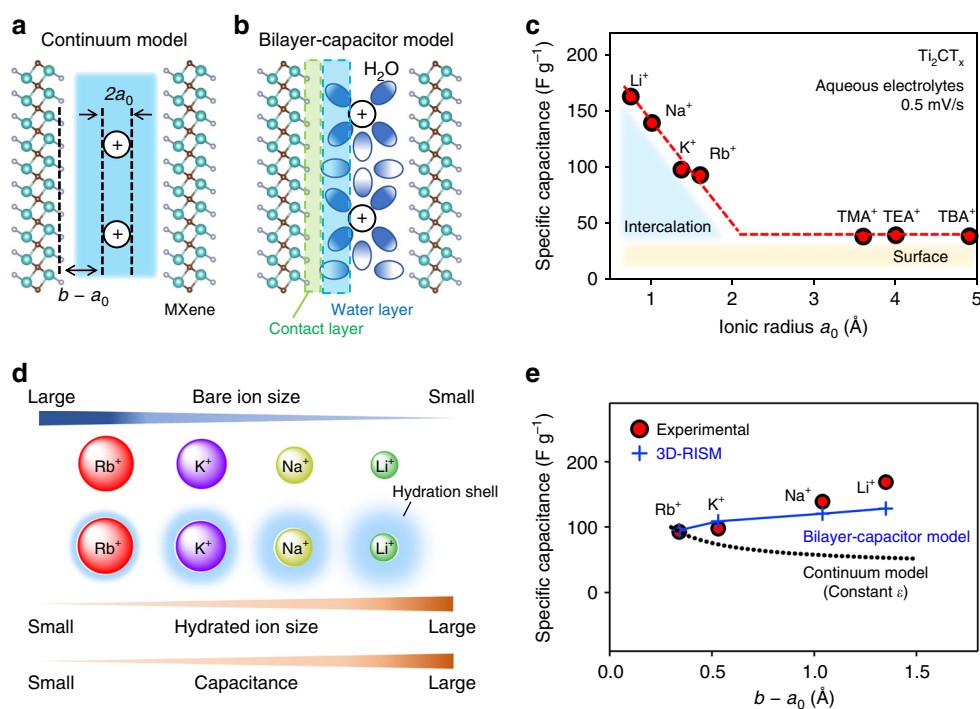

**Fig. 1** Enlargement of the capacitance of a microslit capacitor. **a** Schematic illustration of a continuum model of a microslit capacitor consisting of MXene Ti$_2$CT$_x$ nanosheets. Ti (dark cyan), C (brown), surface termination group T (gray) atoms are shown. **b** Schematic illustration of a bilayer-capacitor model. **c** Experimental specific capacitance of MXene Ti$_2$CT$_x$ with aqueous Li$^+$, Na$^+$, K$^+$, Rb$^+$, TMA$^+$ (tetramethylammonium), TEA$^+$ (tetraethylammonium), and TBA$^+$ (tetrabutylammonium) electrolytes. Each capacitance is calculated from the CV curve at the scan rate of 0.5 mV s$^{-1}$. **d** Orders of bare-ion size, hydrated ion size, and observed capacitance. **e** Ion-MXene distance ($b - a_0$) dependence of the experimental specific capacitance. The black dotted line shows a calculated capacitance based on the continuum model ($C = \frac{\varepsilon_r \varepsilon_0 A}{b - a_0}$ with constant $\varepsilon_r \varepsilon_0$), whereas the blue line shows a calculated capacitance based on the bilayer-capacitor model with variable $\varepsilon_r \varepsilon_0$ from the 3D-RISM calculation

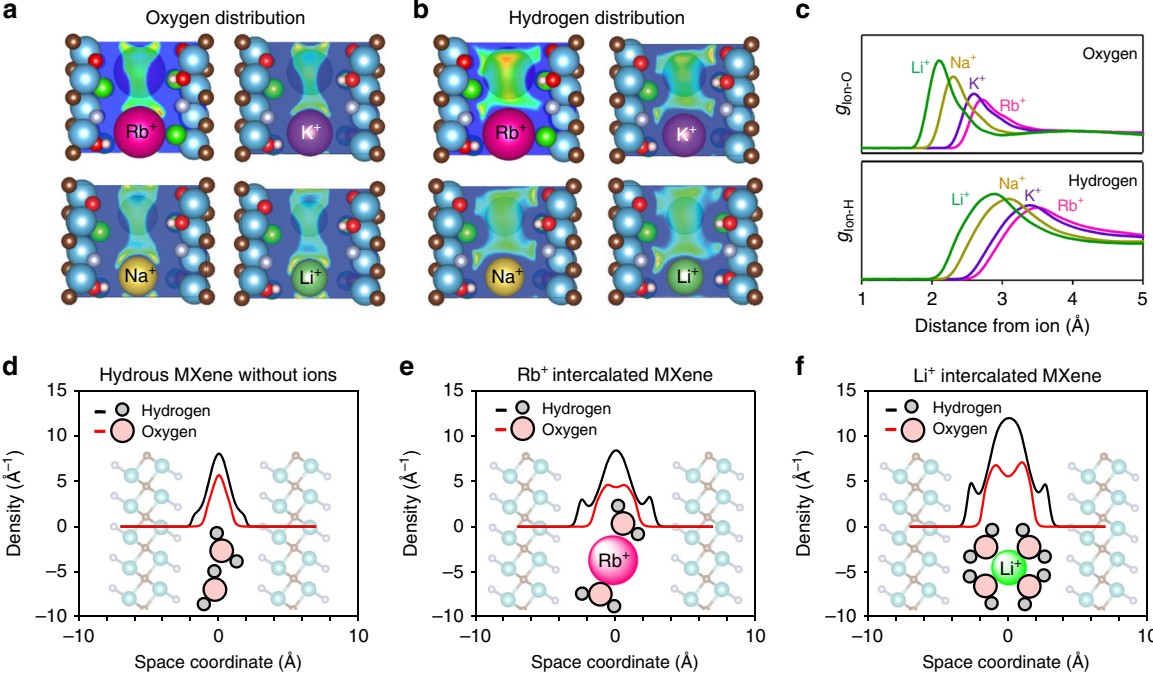

**Fig. 2** 3D-RISM calculation results for the hydrated ions confined in the MXene microslit. **a, b** Oxygen and hydrogen distributions of ion intercalated $Ti_2CT_x{\cdot}nH_2O$. Ti (dark cyan), C (brown), O (red), F (gray), Cl (green), and H (pale pink) atoms are shown. **c** Radial distribution functions of ion–oxygen and ion–hydrogen distances. **d, e, f** Hydrogen and oxygen atomic density profiles along the c axis (perpendicular to the MXene layers) in hydrous, $Rb^+$ intercalated, and $Li^+$ intercalated $Ti_2CT_x{\cdot}nH_2O$. The optimized n values are 0.5, 0.8, and 1.35 for the hydrous, $Rb^+$ intercalated, and $Li^+$ intercalated MXenes, respectively, and these values are consistent with the thermogravimetric experimental results (Supplementary Fig. 14b)

water strongly overscreens an external electric field within the MXene EDLC electrode, leading to an anomalously enhanced capacitance.

## Results

**Intercalation capacitance of MXene.** MXene $Ti_2CT_x$ was synthesized by removal of Al from $Ti_2AlC$ (Supplementary Fig. 1) with LiF-HCl aqueous solution[17]. The chemical composition of the resulting compound after complete dehydration at 200 °C (anhydrous MXene) was determined as $Ti_2C(OH)_{0.3}O_{0.7}F_{0.6}Cl_{0.4}$ using a standard microanalytical technique. As the water molecules can easily penetrate into the MXene interlayer, anhydrous MXene transforms to a hydrous form $Ti_2CT_x{\cdot}0.5H_2O$ (hydrous MXene) in ambient atmosphere. A transmission electron micrograph confirms a stacked structure of the MXene nanosheets (Supplementary Fig. 2), which enables a capacitance from ion intercalation (intercalation capacitance; Supplementary Note 1). The EDLC electrode consisting of the hydrous MXene was fabricated without delaminating the stacked structure of the $Ti_2CT_x$ layers. However, scanning electron micrographs (Supplementary Fig. 3) indicate an existence of partially exfoliated structures allowing a certain contribution to a capacitance from surface ion adsorption (surface capacitance)[18]. To spotlight on the two-dimensional confinement effect on the intercalation capacitance of the MXene nanosheets, we carefully separated the two contributions using various aqueous electrolytes (Fig. 1a–e). First of all, to isolate the contribution from the surface capacitance, we measured capacitances with electrolytes consisting of large alkylammonium cations $[N(C_nH_{2n+1})_4]^+$ (n = 1, 2, and 4) (Supplementary Fig. 4), because these large cations cannot be intercalated into nanoscale space between the MXene nanosheets. We believe this simple methodology provides better analysis than a widely used method using a rate-dependent capacitance (Supplementary Fig. 5)[19]. As a result (Fig. 1c), for all tested alkylammonium

cation species a specific capacitance of the MXene electrode is ~40 F g$^{-1}$, which is attributed to the surface capacitance. This interpretation was validated by ex situ X-ray diffraction (XRD) patterns for the charged electrode, which do not indicate any change in the interlayer distance (Supplementary Fig. 6), excluding the possibility of alkylammonium cation intercalation.

Having determined the surface capacitance of the MXene electrode, we measured the intercalation capacitance using electrolytes consisting of small alkali cations (Li$^+$, Na$^+$, K$^+$, and Rb$^+$) at the scan rate of 0.5 mV s$^{-1}$. The MXene electrode gave a much larger specific capacitance (90–160 F g$^{-1}$) than the surface capacitance (Fig. 1c and Supplementary Fig. 4), and the ex situ XRD patterns indicate an increase in the interlayer distance upon charging (Supplementary Fig. 6). Furthermore, ex situ Ti K-edge spectroscopy suggests the reversible redox of Ti upon charge/discharge (Supplementary Fig. 7). All these experimental observations confirm the occurrence of intercalation capacitance. Cations are intercalated into nanoscale space between the MXene nanosheets, which gives a capacitance (C) of a slit capacitor with pore sizes < 2 nm expressed as follows[13]:

$$C = \frac{\varepsilon_r \varepsilon_0 A}{b - a_0} \qquad (2)$$

where $\varepsilon_r$ is the total dielectric constant between the electrode surface and the ion, $\varepsilon_0$ is the vacuum permittivity, A is the total surface area of both walls, $a_0$ is the ionic radius[20,21], and 2b is the separation of the slit walls (Fig. 1a). It is important to note that, in contrast to a cylinder capacitor (Supplementary Table 1), b for the slit capacitor depends on the alkali ion species intercalated in the slit. This equation predicts that the capacitance of the slit capacitor increases as the effective distance $(b - a_0)$ between the electrode surface and the counterion decreases. However, contrary to the prediction of the above equation (black dotted line, Fig. 1e), the specific capacitance of the MXene electrode

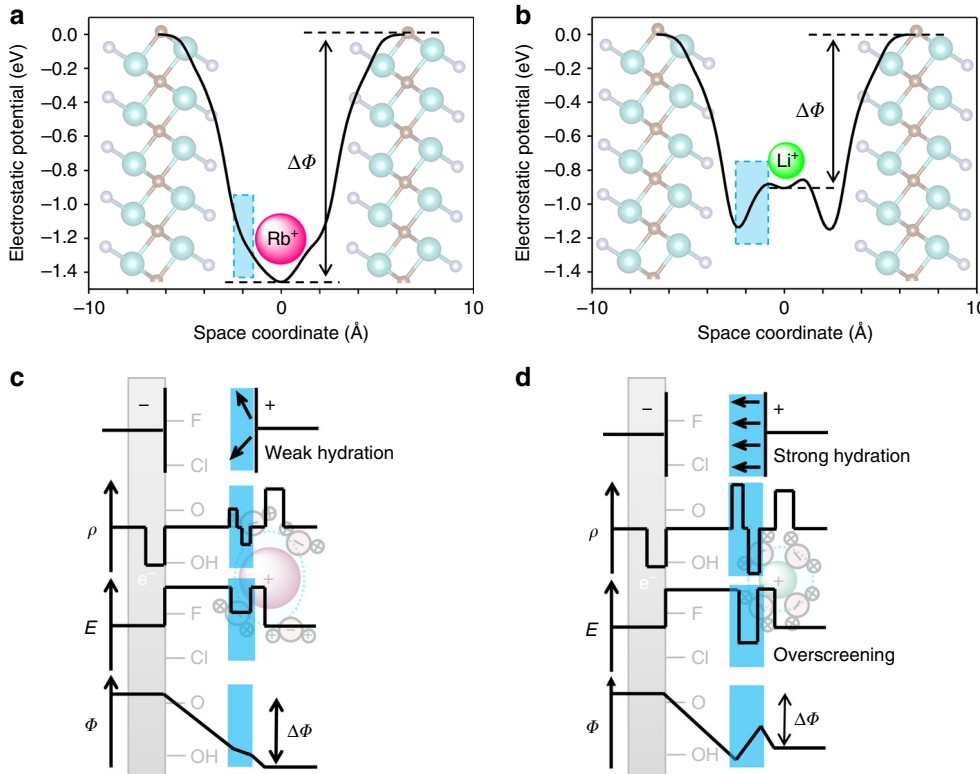

**Fig. 3** Negative dielectric constant of confined water. **a**, **b** Theoretical calculation for the electrostatic potential profile of Rb+- and Li+-intercalated Ti$_2$CT$_x$·$n$H$_2$O. **c**, **d** Schematic illustrations of the charge density ($\rho$), the strength of the electric field ($E$), and the electrostatic potential ($\Phi$) of weakly hydrated cations and strongly hydrated cations confined in a microslit supercapacitor as a function of the spatial coordinates. The arrows indicate the direction of the water dipole moments

increased as $b - a_0$ increases (Li$^+$ > Na$^+$ > K$^+$ > Rb$^+$). The enlarged capacitance, e.g., for a Li$^+$ electrolyte is observed even at faster charge/discharge rates and during 300 charge/discharge cycles (Supplementary Fig. 8). The similar capacitance enlargement is generally observed for other MXenes (Ti$_3$C$_2$T$_x$ and Mo$_2$CT$_x$) (Supplementary Figs. 9 and 10).

The anomalous increase of the MXene electrode capacitance in the order of Li$^+$ > Na$^+$ > K$^+$ > Rb$^+$ is further highlighted when considering that a conventional activated carbon EDLC electrode exhibits a roughly constant capacitance independent of the alkali ions species (Supplementary Figs. 11 and 12). XRD analysis shows that the intercalation of ions with larger "hydrated" ionic radii (Fig. 1d, Li$^+$ > Na$^+$ > K$^+$ > Rb$^+$) gave rise to a larger separation in the slit walls (Supplementary Fig. 6), whereas the $^1$H magic angle spinning (MAS) NMR indicates that the amount of confined water in MXene nanosheets increases after hydrated-ion intercalation (Supplementary Fig. 13). Therefore, water molecules are definitely co-intercalated (Fig. 1b) and we expect that the hydration shell has an important role in determining the structure inside the microslit and thus the anomalous capacitance.

**3D-RISM calculations of MXene.** To further understand and theoretically model the hydration structure in the MXene microslit capacitor for various alkali ions, we conducted a three-dimensional reference interaction site model (3D-RISM) calculation[22–25]. The 3D-RISM calculation allows the simulation of a 3D distribution of solvent molecules, as well as calculations of solvation energy and optimal solvent density (Fig. 2a–f). Before cation intercalation (Fig. 2d), the atomic density profile along the $c$ axis (perpendicular to the MXene layers) suggests that both

oxygen and hydrogen atoms primarily accumulate in the central plane of the microslit (away from the MXene interface). We presume that there is only a weak interaction between water and the MXene surface. Osti et al.[26] also reported that water in pristine Ti$_3$C$_2$T$_x$ has bulk-like characteristics, which indicates the weak interaction between water and MXene.

3D-RISM calculations were then conducted for cation-intercalated Ti$_2$CT$_x$·$n$H$_2$O. The amount of H$_2$O between the MXene layers, $n$, was optimized energetically (Supplementary Fig. 14a) for each alkali ion with the fixed interlayer distance that was experimentally determined using the XRD patterns (Supplementary Fig. 6). The optimized hydration structures indicate the accommodation of larger amounts of water in MXene in the order of Li$^+$ > Na$^+$ > K$^+$ > Rb$^+$. This trend is explained by the increasing hydration energy of cations as their bare ionic radii decrease. The experimentally determined amount of water for each MXene using thermogravimetric (TG) analyses is in perfect agreement with the 3D-RISM calculation results (Supplementary Fig. 14b).

The oxygen distribution in the cation-intercalated MXenes (Fig. 2a) indicates that the oxygen atoms accumulate inside the hydration shells of the cations. Based on the radial distribution function of an ion–oxygen distance (Fig. 2c), the oxygen density immediately around Li$^+$ is much higher than that around Rb$^+$, suggesting the stronger hydration energy of Li$^+$ compared with Rb$^+$. Indeed, the oxygen density profile in the Li$^+$-intercalated MXene along the $c$ axis contains two peaks around Li$^+$ and these peaks are more intense than those around Rb$^+$ (Fig. 2e, f). Simultaneously, the hydrogen density profiles for the cation-intercalated MXene (Fig. 2b) indicate considerable hydrogen density close to the MXene surface. Considering a small hydrogen density near MXene before cation intercalation, the hydrogen

atoms of water distributed along the surface of each MXene layer are predominantly due to the formation of the hydration shell around intercalated cations (Fig. 2e, f). The 3D-RISM calculations for a series of alkali ions show that the cations with smaller bare ionic radius tend to exhibit a larger hydrogen density near the MXene surface (Supplementary Fig. 15). The significant interference to water arrangement between $Ti_3C_2T_x$ nanosheets by cation intercalation was also observed by Osti et al.[26]. It is most likely to be that the hard hydration shell of strong Lewis acid cations such as $Li^+$ forces the hydrogen atoms to reside close to the surface of the MXene layer, whereas the soft hydration shell of weak Lewis acid cations such as $Rb^+$ deforms easily when confined within the microslit.

After confirming that the hydrogen and oxygen distributions in the microslit depend on the intercalated ionic species, we calculated the electrostatic potential profiles, relative to the MXene electrode (Fig. 3a, b and Supplementary Fig. 16). In the $Rb^+$-intercalated MXene (Fig. 3a), the electrostatic potential ($\Phi$) monotonically decreases from the MXene layer until very close to the ion location in the center of the microslit, as expected for a conventional capacitor sandwiching a dielectric layer. The charge density ($\rho$), electric field ($E$), and $\Phi$ are depicted schematically in Fig. 3c. For $Li^+$-intercalated MXene (Fig. 3b), in contrast, the profile change in $\Phi$ is not monotonic; $\Phi$ rises near the locus of $Li^+$, leading to a reduced total potential difference ($\Delta\Phi$). The 3D-RISM calculations for intercalation of a series of alkali ions show that the smaller cation-intercalated MXenes induces more reduced total $\Delta\Phi$ (Supplementary Fig. 16). Considering the capacitance ($C$) is expressed as,

$$C = \frac{\Delta Q}{\Delta\Phi} \qquad (3)$$

where $\Delta Q$ is the stored charge by applying $\Delta\Phi$ (Supplementary Note 1), the reduced $\Delta\Phi$ explains the larger capacitance of the MXene electrode in the order of $Li^+ > Na^+ > K^+ > Rb^+$ (Fig. 1d). Indeed, the calculated $C$ based on the value of $\Delta\Phi$ from 3D-RISM well reproduces the experimental $C$ (blue line, Fig. 1e and Supplementary Fig. 17).

## Discussion

Then, we consider the origin and implications of the reduced $\Delta\Phi$ specifically observed in $Li^+$-intercalated MXene. The electric-double layer formed in MXene system can be modeled as an equivalent circuit of a bilayer capacitor (Fig. 1b). The capacitance of a capacitor sandwiching two series of a contact layer and a water layer can be expressed as,

$$C = \frac{A\varepsilon_0}{\lambda} \qquad (4)$$

where

$$\lambda = \frac{l^c}{\varepsilon_r^c} + \frac{l^h}{\varepsilon_r^h} \qquad (5)$$

with thicknesses ($l^c$, $l^h$) and dielectric constants ($\varepsilon_r^c$, $\varepsilon_r^h$) for a contact layer and a water layer, respectively[27].

Based on density functional theory (DFT) calculations[28], in the first term $\frac{l^c}{\varepsilon_r^c}$ for the contact layer between the water and an electrode, $\varepsilon_r^c$ is small to be around $10^0$ and $l^c$ is 2–3 Å. In the second term $\frac{l^h}{\varepsilon_r^h}$ for the water layer confined in spaces ranging from macroslit ($2b > 50$ nm) to mesoslit ($2$ nm $< 2b < 50$ nm) (Supplementary Table 1), water molecules are weakly bounded to rotate freely and give a large positive $\varepsilon_r^h$ of ~80, where the situation becomes as such $\frac{l^c}{\varepsilon_r^c} \gg \frac{l^h}{\varepsilon_r^h}$, and hence $\lambda \sim \frac{l^c}{\varepsilon_r^c}$[28].

Therefore, in macroslit and mesoslit capacitors, the dielectric contact layer (the first term) dominates the overall capacitance, where the water molecules do not have essential roles.

In striking contrast, in a microslit capacitor, the water layer (the second term) largely influences the overall capacitance. The increase in $\Phi$ near the locus of $Li^+$ indicates excess polarization (called overscreening)[29,30] and inversion of $E$ at the confined water layer (i.e., hydration shell) relative to the external electric field ($E_{ext}$; Fig. 3d). This inversion of $E$ $\left(\text{where } E = \frac{E_{ext}}{\varepsilon_r}\right)$ indicates that the dielectric constant of the hydration shell is negative. As theoretically shown by Bopp et al.[29], a negative dielectric constant of water is possible under the condition of microscopic overscreening, which is realized by a resonant effect between dipolar polarization of water and an external electric field. Resonant conditions were suggested as follows: (i) an external electric field has several Å modulation and (ii) water is confined between a slit wall and an ion[30].

For a microslit ($2b < 2$ nm) capacitor, such as the MXene system described here, we presume that the resonant condition (i) on double-layer thickness is satisfied for all of $Li^+$, $Na^+$, $K^+$, and $Rb^+$ ions based on calculation results (Supplementary Fig. 16), whereas the resonant condition (ii) on firm water confinement is satisfied only by strong Lewis acid cations such as $Li^+$. The soft hydration shell of the weakly hydrated cations such as $Rb^+$ largely deforms even in the microslit as demonstrated in Fig. 2e, whereby water molecules are not confined effectively, and remain to have a positive dielectric constant. In contrast, the hard hydration shell of $Li^+$ is strongly confined between the MXene wall and $Li^+$ as evidenced in Fig. 2f, and water dipolar polarization resonantly overscreens the external electric field to induce an inverse $E$. It is this situation that can lead to a negative dielectric constant for the hydration layer and hence an increase in capacitance. Importantly, the overscreening of a hydration shell confined in a microslit capacitor is a general phenomenon: e.g., the 3D-RISM calculation for $Li^+$-intercalated $Mo_2CT_x$ also indicates the negative dielectric constant of confined water (Supplementary Fig. 18), explaining the capacitance enhancement experimentally observed for $Mo_2CT_x$ (Supplementary Fig. 10). As the simplest combination, the water confined in the microslit consisting of graphene is also predicted to exhibit the negative dielectric constant and the overscreening behavior (Supplementary Fig. 19). Furthermore, Geng et al.[31] reported that the capacitance of metallic 1T $MoS_2$ with an aqueous $Li^+$ electrolyte is larger than that with an aqueous $Na^+$ electrolyte. These verifications strongly suggest that exploiting the water confinement effect is a versatile strategy to enhance the capacitance of microslit capacitors.

In general, the strategy of using larger surface area materials to increase the gravimetric capacitance severely suffers from smaller electrode density and smaller volumetric capacitance thereof[32]. Our discovery of a negative dielectric constant of confined water and its contribution to larger capacitance of the microslit capacitor not only solve this long-standing dilemma but also offer an important prospect that stacked two-dimensional materials might have considerable potential as EDLC electrodes. The capacitance enlargement by confined negative dielectric water could be extended to other systems. For example, $H^+$ intercalation, which was not considered in this work, has been reported to exhibit a large capacitance through the protonation of MXene[33]; therefore, the influence of water to the $H^+$ capacitance would be of particular interest to further increase the capacitance. The influence of non-aqueous electrolyte solvents to the MXene capacitance is also an important issue that needs to be clarified (Supplementary Fig. 20), as it includes more complicated phenomena such as interfacial desolvation, solid-electrolyte interphase formation, and co-intercalation[34,35]. Moreover, existence of water molecules with a negative dielectric constant casts doubt on a conventional

capacitor model postulated by Stern[36] with low-dielectric surface water layer and potentially leads to redefinition of how to construct theoretical models for EDLC electrodes. Further studies on similar confinement effects in other microporous materials are expected to stimulate a range of striking discoveries of EDLCs with higher gravimetric and volumetric energy densities.

## Methods

**Synthesis of MXene**. $Ti_2AlC$ was prepared by high-frequency induction heating of a precursor mixture consisting of TiC, Ti, and Al at 1300 °C for 1 h under an Ar flow. $Ti_2CT_x$ was synthesized by reacting 0.5 g of $Ti_2AlC$ powder with an aqueous mixture of 2 M LiF and 6 M HCl for 12 h at room temperature. The treated powder was dried under vacuum at 200 °C for 24 h (anhydrous MXene). The chemical composition of anhydrous MXene $Ti_2CT_x$ was determined by the standard microanalytical method for C, H, F, and Cl, and by X-ray photoelectron spectroscopy for an O/OH ratio. Calc. (Found) for $Ti_2C(OH)_{0.3}O_{0.7}F_{0.6}Cl_{0.4}$: C: 8.03% (7.50%), H: 0.20% (0.25%), F: 7.62% (7.84%), Cl: 9.48% (9.22%). $Ti_3C_2T_x$ and $Mo_2CT_x$ were synthesized according to the procedures reported previously[5,37].

**Characterization**. The powder XRD patterns were recorded on a Rigaku SMART-LAB powder diffractometer with Cu Kα radiation with a step of 0.02° over a 2θ range of 3°–80°. Samples for ex situ XRD patterns were prepared electrochemically and were used for the measurements without drying. The number of intercalated water molecules in MXene was estimated by TG analysis. The TG data were collected on a Seiko Extar 6000 TG/DTA instrument over a 30–400 °C temperature range using an Ar gas atmosphere. The heating rate was fixed at 5 K min$^{-1}$.

**Electrochemistry**. For the electrochemical measurements, the working electrode was fabricated by mixing $Ti_2CT_x$, acetylene black, and polytetrafluoroethylene in 80:10:10 weight ratio. The resulting paste was pressed onto a nickel mesh. A three-electrode glass cell was assembled with a Pt mesh as the counter electrode and Ag/AgCl in saturated aqueous solution of KCl for a reference electrode. Aqueous solutions (0.5 M) of $Li_2SO_4$, $Na_2SO_4$, $K_2SO_4$, $Rb_2SO_4$, tetramethylammonium chloride, tetraethylammonium chloride, and tetrabutylammonium chloride were used as the electrolytes. The sweep rate of the cyclic voltammetry (CV) measurements was set to 0.5 and 2.0 mV s$^{-1}$, and the cutoff voltages were −0.7 V and −0.2 V (vs. Ag/AgCl). The specific capacitance from the CV curve was calculated as $\frac{1}{\Delta V}\int \frac{j(V)}{s}dV$, where $V$ is the potential, $\Delta V$ is the potential window, $j(V)$ is the specific current, and $s$ is the scan rate. The activated carbon for the reference experiment (Supplementary Figs. 6 and 7) was purchased from Kansai Coke and Chemicals (MSC-30 with a specific surface area of 3000 m$^2$ g$^{-1}$). The X-ray absorption spectra were measured in the transmission mode at room temperature at BL-9C of Photon Factory. The X-ray energy for each edge was calibrated by using a corresponding metal foil. The obtained experimental data were analyzed using Rigaku REX2000 software. The $^1H$ MAS NMR spectra were recorded at frequency of 800 MHz (18.79 T) using a JEOL JNM-ECA800 system equipped with a JEOL 1.0 mm HXMAS probe. To reduce the $^1H$ background signal from the probe material, the DEPTH2 pulse sequence was used. The experimental conditions were set up with 90° pulse length of 1.2 μs, recycle delay of 5 s, and the MAS rate of 60 kHz. The $^1H$ chemical shift was referenced to the peak of silicon rubber and set to 0.12 p.p.m. from tetramethylsilane.

**Calculations**. The hydration energy and atomic distributions are calculated by using a 3D-RISM theory combined with DFT. The 3D-RISM code is implemented into a DFT simulation package named "Quantum Espresso"[24,38]. The exchange–correlation functional was used within the generalized gradient approximation proposed by Perdew et al.[25]. The ultrasoft pseudopotential scheme combined with plane-wave basis sets imposing cutoff energies of 30 and 300 Ry was used to describe the Kohn–Sham orbitals and electron density, respectively. The Brillouin-zone summation was evaluated using a $3 \times 3 \times 1$ k-point grid for structure optimization and total energy calculation. The convergence criteria for structure optimization included $10 \times 10^{-3}$ Ry per Bohr for forces and $10 \times 10^{-4}$ Ry for the energy.

The structural model of MXene $Ti_2CT_x$ consisted of $2 \times 2\sqrt{3}$ rectangular supercell with four different surface functions (representing F, Cl, O, and OH). The chemical composition of hydrous MXene was assumed to be $Ti_2C$ $(F_{0.5}Cl_{0.5}O_{0.5}OH_{0.5}) \cdot 0.5H_2O$. The lattice constants of the supercell were theoretically optimized as $12.326 \times 21.349$ Å$^2$. The interlayer distance between the central carbon layers of the adjacent MXene sheets was set to experimentally determined values (i.e., 13.2, 13.1, 12.8, 12.7 Å for $Li^+$-, $Na^+$-, $K^+$-, and $Rb^+$-intercalated models, respectively). Each intercalated model has two ions located at (1/4, 1/4, 1/2) and (3/4, 3/4, 1/2) in the fractional coordinates of each supercell. The validity of our structural models was confirmed by the perfect agreement between the experimentally determined and the theoretically optimized water contents ($n$) in various cation-intercalated MXenes (Supplementary Fig. 14). The calculated capacitance ($C$) for each cation-intercalated MXene (inset in Fig. 1b) was obtained

by the equation as,

$$C = C_{surface} + \frac{\Delta Q_{intercalation}}{\Delta \Phi_{calc}} \tag{6}$$

where $C_{surface} = 40$ F g$^{-1}$, $\Delta Q_{intercalation} = 80.6$ C g$^{-1}$ (0.125 cation intercalation per the formula unit), $\Delta \Phi(Rb)_{calc} = 1.46$ eV, $\Delta \Phi(K)_{calc} = 1.17$ eV, $\Delta \Phi(Na)_{calc} = 1.0$ eV, and $\Delta \Phi(Li)_{calc} = 0.91$ eV, respectively.

## Data availability

The whole datasets are available from the corresponding author on request.

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

## Acknowledgements

This work was financially supported by the Ministry of Education, Culture, Sports, Science and Technology (MEXT), Japan under the "Elemental Strategy Initiative for Catalysts and Batteries (ESICB)." This work was also supported by MEXT, Japan, and Grant-in-Aid for Specially Promoted Research Number 15H05701. M. Okubo was financially supported by JSPS KEKENHI Grant Numbers JP15H03873, JP16H00901, and 18H03924. We are grateful to Satomichi Nishihara for implementation of the 3D-RISM into Quantum Espresso package. X-ray absorption spectroscopy was conducted under the approval of the Photon Factory Program Advisory Committee (Proposal 2016G031 and 2018G082).

## Author contributions

M. Okubo and A.Y. conceived and directed the project. A.S. and S.K. synthesized and characterized MXenes. A.S. measured and analyzed the electrochemical properties. A.S., K.Y., and K.G. conducted and analyzed the $^1$H MAS NMR spectra. Y.A. and M. Otani conducted the 3D-RISM calculations. All authors wrote the manuscript.

## Additional information

**Competing interests:** The authors declare no competing interests.

