## [Peer Review File · Nature Communications]

Reviewers' comments:

Reviewer #1 (Remarks to the Author):

The current revised version of this manuscript is surely improved but still not to a satisfactory level that fully warrants its publication. The idea of using an exotic phrase like "negative dielectric constant of confined water" is eye-catching, however, to support this concept both experimental evidences and discussion must be convincing and complete. The lack of direct method to measure the dielectric constant is not compensated by the proposed theoretical model, which, in its current format, confuses the reviewer and gives an impression of over-simplification. More significant data and analysis are needed before can be considered for acceptance. Below are the detailed points for the authors to address and further improve the overall quality of the manuscript.

1. I think the authors misunderstood the Comment 1 from Reviewer 1, as the reviewer concerns if the nature of the intercalation capacitance to be EDLC or battery type (DOI: 10.1038/ncomms12647). I didn't see any problem using Bruce Dunn's method in this case, especially when the authors were able to collect rate capabilities.

2. Two of the reviewers emphasized the importance to include experimental and/or theoretical input for the cases in organic electrolytes, however, the authors seemed reluctant to do so. Instead, the authors provided a simple comparison in CV and concluded a negligible capacitance, which was presumably due to "the large energy barrier for desolvation at interface." I highly recommend the authors to seriously consider this point and elaborate their arguments more rigorously. Substantial capacitance of MXene in organic electrolytes is not something unheard of, even for the same material used in this study (J. Power Sources, 2016, 306, 510). And the process of intercalation in organic electrolyte is more complex way beyond merely desolvation. The authors are suggested to contrast the case of organic vs. aqueous electrolytes in maintext considering multiple aspects, such as desolvation, effective ionic radius, solvent co-intercalation, etc.). Any proposed significant difference between the two systems must be verified experimentally or theoretically. Figure R2 should go to SI with more experimental details (e.g. scan rate is missing).

3. The inclusion of the microslit consisting of graphene beyond MXene is plausible but also arbitrary. First, I don't understand the reason why the separation of graphene sheets is assumed at 12 Å. Second, can the authors provide any general design/selection rule in order for this "negative dielectric constant" concept to be applied to a broader range of material systems?

4. The authors denied many allegations from Reviewer 2, for example the ones related to the ionic radius. The reviewer may have been misled by Figure 1b where naked ionic radius was used, where in responses the authors emphasized heavily on hydrated ion radius. If it is that important, then why didn't the authors try to generate a correlation between capacitance and hydrated ionic size? This problem also appears in Comment 4 from Reviewer 3. Notice the Stokes radii are much smaller than the interlayer distances, some significant changes are expected to exist for the hydration shell. More elaboration at "molecular-level" is needed to describe the dynamic process from solvated ions in the solution to water confined ions in the microslit. If the focus of this study is the "discovery of negative dielectric constant of confined water in nanosheets, which can enhance a capacitance of a capacitor electrode", I would expect a capacitance vs. a quantity related to the dielectric constant, instead of ionic radius.

5. More questions can be asked in terms of the ionic radius. No matter in the basic model or the 3D-RISM model, the effect of the hydrated-ion size on the ion packing density inside the material (correspondingly the capacitance) is not considered. Since different ions were used and they have

different hydrated ionic radii, the authors need to take their arrangement and interionic interaction into consideration.

6. I agree the case of H⁺ is somewhat different from alkali cations. However, the reference brought up by the authors (ACS Nano 2016, 10, 11344) actually manifests the complex nature of intercalation process any many aspects that was not discussed in this work (such as ion mobility and the role of counterion in charge storage mechanism). I suggest to add this reference in the manuscript mentioning this aspect at a proper place.

Reviewer #2 (Remarks to the Author):

The authors addressed the issues carefully, but some problems still existed. They claimed that the Dunn's method was not suitable to analyze the capacitive behavior of 2D MXene-based electrodes in their responsive letter. More description should be added, especially the difference of assumption between linear diffusion and radical diffusion. Although the authors extended the concept to another type of 2D MXene-Mo₂CTx, other 2D materials should be more useful to support the finding of this work.

Reviewer #3 (Remarks to the Author):

This work focuses on the dielectric constant of electrolyte in the study of supercapacitors and reports the negative dielectric constant of water in a confined slit-shaped space. This study together with the new results is demonstrated the importance for the understanding of supercapacitors and new design of high performance devices particularly for highly volumetric performance ones since the narrowly slit-shaped space is dominant for the reported effect. Considering the new insight of supercapacitors which is supported by some interesting discussion, the referee recommend this submission for publication with some concerns that should be addresses.

- 1, the authors should give more distinct definitions for negative dielectric constant of water, the physical nature, and the range of the function, etc.
- 2, the authors is suggested to give a typical example for the design of novel devices highlighting both the gravimetric and volumetric performances, or at least give a more detailed comments on how to use the new finding to design new devices.
- 3, please give some discussion what types of confined space is needed for this negative dielectric constant, which must be slit-shaped or not?
- 4, since this study add new insights for the design of volumetric devices design, please cite and comment some important references for volumetric supercapacitor study for supercapacitors, like Energy Environ. Sci., 2015, 8, 1390-1403.
- 5, the references should be formatted (e.g. ref. 7, last and first names should be corrected).

Reviewer #1 (Remarks to the Author):

- (1) The current revised version of this manuscript is surely improved but still not to a satisfactory level that fully warrants its publication. The idea of using an exotic phrase like “negative dielectric constant of confined water” is eye-catching, however, to support this concept both experimental evidences and discussion must be convincing and complete. The lack of direct method to measure the dielectric constant is not compensated by the proposed theoretical model, which, in its current format, confuses the reviewer and gives an impression of over-simplification. More significant data and analysis are needed before can be considered for acceptance. Below are the detailed points for the authors to address and further improve the overall quality of the manuscript.

Author reply: We appreciate valuable suggestions.

- (2) I think the authors misunderstood the Comment 1 from Reviewer 1, as the reviewer concerns if the nature of the intercalation capacitance to be EDLC or battery type (DOI: 10.1038/ncomms12647). I didn't see any problem using Bruce Dunn's method in this case, especially when the authors were able to collect rate capabilities.

Author reply: As the reviewers #1 and #2 required, we have added the results obtained from the Dunn's method in the supplementary Figure S5. However, as explained in this Figure, the assumptions of this method are incorrect for the composite electrode of layered MXene. We have added the explanations in the main text (P5, L13) and in the supporting information with the suggested reference (Lukatskaya *et al.*, *Nat. Commun.* 2016) as,

‘First of all, in order to isolate the contribution from the surface capacitance, we measured capacitances with electrolytes consisting of large alkylammonium cations $[N(C_nH_{2n+1})_4]^+$ ($n = 1,$

2, and 4) (Supplementary Fig. S4), because these large cations cannot be intercalated into nanoscale space between the MXene nanosheets. We believe this simple methodology provides better analysis than a widely-used method using a rate-dependent capacitance (Supplementary Fig. S5).^{19,}

(a)

$i = Av^{0.5} + Bv$ (theory) $i = Av^b$ ($0.5 < b < 1.0$) (empirical) V. Augustyn, et al. & B. Dunn, Nat. Mater. (2013) 12, 518.	Applicability to MXene (capacitive behavior, composite electrode, layered structure)
Assumption 1: Fick's equation ($\text{grad}(\Phi)=0$)	Incorrect (Poisson-Nernst-Planck equation must be used)
Assumption 2: Dominant diffusion polarization of an active material	Incorrect (other possible rate-determining steps such as ion migration in a composite and IR drops)
Assumption 3: linear diffusion (1D)	Incorrect (radial diffusion (2D))

(b)

Figure S5. (a) Assumptions in the capacitance vs. diffusion evaluation method proposed by V. Augustyn, et al. (*Nat. Mater.* **2013**, 12, 518) and its applicability to MXene. (b) Capacitance evaluation using the rate-dependent capacitance ($i = Av^{0.5} + Bv$) of Ti_2CT_x . The rate-dependent current i of MXene electrodes is governed by the Poisson-Nernst-Planck (PNP) equation

$$\frac{\partial C}{\partial t} = D\Delta C + \frac{FD}{RT} \text{grad } C \cdot \text{grad } \Phi + \frac{FD}{RT} C \Delta \Phi$$

under the initial and boundary conditions of a cylinder electrode. Here, C , t , D , F , R , T , and Φ are ion concentration, time, diffusion coefficient, Faraday constant, gas constant, temperature, and inner potential. However, as the PNP equation cannot be solved analytically, the widely-used rate dependence for diffusion current ($i = Av^{0.5}$) has no theoretical background. See, for example, K. Aoki, *Electroanalysis* **5**, 627 (1993).

(3) Two of the reviewers emphasized the importance to include experimental and/or theoretical input for the cases in organic electrolytes, however, the authors seemed reluctant to do so. Instead, the authors provided a simple comparison in CV and concluded a negligible capacitance, which was presumably due to “the large energy barrier for desolvation at interface.” I highly recommend the authors to seriously consider this point and elaborate their arguments more rigorously. Substantial capacitance of MXene in organic electrolytes is not something unheard of, even for the same material used in this study (*J. Power Sources*, 2016, 306, 510). And the process of intercalation in organic electrolyte is more complex way beyond merely desolvation. The authors are suggested to contrast the case of organic vs. aqueous electrolytes in main text considering multiple aspects, such as desolvation, effective ionic radius, solvent co-intercalation, etc.). Any proposed significant difference between the two systems must be verified experimentally or theoretically. Figure R2 should go to SI with more experimental details (e.g. scan rate is missing).

Author reply: We have added the comparison of the CV curves with aqueous and nonaqueous electrolytes in the supporting information. With no doubt, it is important to study the influence of nonaqueous electrolyte solvents to the capacitance, however, as suggested by the reviewer, we should consider many factors including desolvation at interface, SEI formation (and associated irreversibility), and co-intercalation. Indeed, by the authors themselves, complicated and irreversible lithiation was already reported for a Ti_2CT_x electrode charged with 1 M $\text{LiPF}_6/\text{EC-DMC}$ electrolyte (M. Okubo *et al.*, *Acc. Chem. Res.* (2018) 51, 591). At present with major focus on peculiar features with aqueous electrolyte, detailed aspects upon use of organic electrolytes are out of the scope of this paper and further insights will be reported elsewhere. We have commented on this perspective in the conclusion section with the suggested reference (Y. Dall’Agnese, *et al.*, *J. Power Sources*, 2016) in P12, L10 as,

‘The influence of nonaqueous electrolyte solvents to the MXene capacitance is also an important issue to be clarified (Supplementary Fig. S20), as it includes more complicated phenomena such as interfacial desolvation, SEI formation, and co-intercalation.’^{34,35}

- (4) The inclusion of the microslit consisting of graphene beyond MXene is plausible but also arbitrary. First, I don't understand the reason why the separation of graphene sheets is assumed at 12 Å. Second, can the authors provide any general design/selection rule in order for this “negative dielectric constant” concept to be applied to a broader range of material systems?

Author reply: We have carried out the 3D-RISM calculations of graphene with hypothetical various separations, and found the negative dielectric constant at the separation of 12 Å. However, the details of these calculations on graphene system are out of the scope of this paper on MXene and will be published elsewhere.

As for the general criteria for negative dielectric constant, we have added the detailed explanations in the main text in P10, L2 and P12, L 16 as,

‘The electric-double layer formed in MXene system can be modeled as an equivalent circuit of a bilayer capacitor (Fig. 1b). The capacitance of a capacitor sandwiching two series of a contact layer and a water layer can be expressed as $C = \frac{A\epsilon_0}{\lambda}$, where $\lambda = \frac{l^c}{\epsilon_r^c} + \frac{l^h}{\epsilon_r^h}$ with thicknesses (l^c , l^h) and dielectric constants (ϵ_r^c , ϵ_r^h) for a contact layer and a water layer, respectively.²⁸ Based on DFT calculations²⁹, in the first term $\frac{l^c}{\epsilon_r^c}$ for the contact layer between the water and an electrode, ϵ_r^c is small to be around 10^0 and l^c is 2–3 Å. In the second term $\frac{l^h}{\epsilon_r^h}$ for the water layer confined in spaces ranging from macroslit ($2b > 50$ nm) to mesoslit ($2 \text{ nm} < 2b < 50$ nm) (Supplementary Table S1), water molecules are weakly bounded to rotate freely and give a large positive ϵ_r^h of approximately 80, where the situation becomes as such $\frac{l^c}{\epsilon_r^c} \gg \frac{l^h}{\epsilon_r^h}$, and hence $\lambda \sim \frac{l^c}{\epsilon_r^c}$.²⁹ Therefore, in macroslit and mesoslit capacitors, the dielectric contact layer (the first term) dominates the overall capacitance, where the water molecules do not play essential roles. In striking contrast, in a microslit capacitor, the water layer (the second term) largely influences the overall capacitance.’

‘Further studies on similar confinement effects in other microporous materials are expected to stimulate a range of striking discoveries of EDLCs with higher gravimetric and volumetric energy densities.’

(5) The authors denied many allegations from Reviewer 2, for example the ones related to the ionic radius. The reviewer may have been misled by Figure 1b where naked ionic radius was used, where in responses the authors emphasized heavily on hydrated ion radius. If it is that important, then why didn't the authors try to generate a correlation between capacitance and hydrated ionic size? This problem also appears in Comment 4 from Reviewer 3. Notice the Stokes radii are much smaller than the interlayer distances, some significant changes are expected to exist for the hydration shell. More elaboration at “molecular-level” is needed to describe the dynamic process from solvated ions in the solution to water confined ions in the microslit. If the focus of this study is the “discovery of negative dielectric constant of confined water in nanosheets, which can enhance a capacitance of a capacitor electrode”, I would expect a capacitance vs. a quantity related to the dielectric constant, instead of ionic radius.

Author reply: Thank you for a kind suggestion. According to the reviewer's suggestion, we have summarized the relationship in more intuitive manner between ion sizes and observed capacitances as Fig. 1d and supplementary Scheme S1 not to mislead readers. Also, following the reviewer's request, we have added a capacitance vs. inner potential difference (related to the dielectric constant of water through $\lambda = \frac{l^c}{\epsilon_r^c} + \frac{l^h}{\epsilon_r^h}$) in the supporting information as Figure S17.

Figure S17. Intercalation capacitance of Ti_2CT_x with various aqueous electrolytes as a function of the inner potential difference calculated by 3D-RISM. The inner potential difference is related to the dielectric constant of water ($\lambda = \frac{l^c}{\epsilon_r^c} + \frac{l^h}{\epsilon_r^h}$). The broken line is the calculation result based on

$C = \frac{\Delta Q_{\text{intercalation}}}{\Delta \Phi_{\text{calc}}}$, where $\Delta Q_{\text{intercalation}} = 80.6 \text{ C/g}$ (0.125 cation intercalation per the formula unit).

We have also added the explanation of this figure in the main text (P9, L19) as,

‘Indeed, the calculated C based on $\Delta\Phi$ from 3D-RISM well reproduces the experimental C (blue line, Fig. 1e and supplementary Fig. S17).’

- (6) More questions can be asked in terms of the ionic radius. No matter in the basic model or the 3D-RISM model, the effect of the hydrated-ion size on the ion packing density inside the material (correspondingly the capacitance) is not considered. Since different ions were used and they have different hydrated ionic radii, the authors need to take their arrangement and interionic interaction into consideration.

Author reply: The 3D-RISM calculations include such self-interaction and modified hydration structure of each ion upon packing. The resulting inner potential difference well reproduces the experimentally observed capacitance.

- (7) I agree the case of H^+ is somewhat different from alkali cations. However, the reference brought up by the authors (ACS Nano 2016, 10, 11344) actually manifests the complex nature of intercalation process any many aspects that was not discussed in this work (such as ion mobility and the role of counter ion in charge storage mechanism). I suggest to add this reference in the manuscript mentioning this aspect at a proper place.

Author reply: According to the reviewer’s comment, we have added a comment on H^+ intercalation with the suggested reference in P12, L8 as,

“ H^+ intercalation, which was not considered in this work, has been reported to exhibit a large capacitance through the protonation of MXene, thereby the influence of water to the H^+ capacitance would be of particular interest to further increase the capacitance.³³”

Reviewer #2 (Remarks to the Author):

- (1) The authors addressed the issues carefully, but some problems still existed. They claimed that the Dunn's method was not suitable to analyze the capacitive behavior of 2D MXene-based electrodes in their responsive letter. More description should be added, especially the difference of assumption between linear diffusion and radical diffusion.

Author reply: As the reviewers #1 and #2 required, we have added the results obtained from the Dunn's method in the supplementary Figure S5. However, as explained in this Figure, the assumptions of this method are incorrect for the composite electrode of layered MXene. We have added the explanations in the main text (P5, L12) and in the supporting information with the suggested reference (Lukatskaya *et al.*, *Nat. Commun.* 2016) as,

'First of all, in order to isolate the contribution from the surface capacitance, we measured capacitances with electrolytes consisting of large alkylammonium cations $[\text{N}(\text{C}_n\text{H}_{2n+1})_4]^+$ ($n = 1, 2, \text{ and } 4$) (Supplementary Fig. S4), because these large cations cannot be intercalated into nanoscale space between the MXene nanosheets. We believe this simple methodology provides better analysis than a widely-used method using a rate-dependent capacitance (Supplementary Fig. S5).¹⁹,

(a)

$i = Av^{0.5} + Bv$ (theory) $i = Av^b$ ($0.5 < b < 1.0$) (empirical) V. Augustyn, et al. & B. Dunn, Nat. Mater. (2013) 12, 518.	Applicability to MXene (capacitive behavior, composite electrode, layered structure)
Assumption 1: Fick's equation ($\text{grad}(\Phi)=0$)	Incorrect (Poisson-Nernst-Planck equation must be used)
Assumption 2: Dominant diffusion polarization of an active material	Incorrect (other possible rate-determining steps such as ion migration in a composite and IR drops)
Assumption 3: linear diffusion (1D)	Incorrect (radial diffusion (2D))

(b)

Figure S4. (a) Assumptions in the capacitance vs. diffusion evaluation method proposed by V. Augustyn, *et al.* (*Nat. Mater.* **2013**, 12, 518) and its applicability to MXene. (b) Capacitance evaluation using the rate-dependent capacitance ($i = Av^{0.5} + Bv$) of Ti_2CT_x . The rate-dependent current i of MXene electrodes is governed by the Poisson-Nernst-Planck (PNP) equation $\frac{\partial C}{\partial t} = D\Delta C + \frac{FD}{RT} \text{grad } C \cdot \text{grad } \Phi + \frac{FD}{RT} C \Delta \Phi$ under the initial and boundary conditions of a cylinder electrode. Here, C , t , D , F , R , T , and Φ are ion concentration, time, diffusion coefficient, Faraday constant, gas constant, temperature, and inner potential. However, as the PNP equation cannot be solved analytically, the widely-used rate dependence for diffusion current ($i = Av^{0.5}$) has no theoretical background. See, for example, K. Aoki, *Electroanalysis* **5**, 627 (1993).

(2) Although the authors extended the concept to another type of 2D MXene-Mo₂CT_x, other 2D materials should be more useful to support the finding of this work.

Author reply: Extending the concept of this work to other systems is under progress and will be

reported elsewhere. We have added comments on this perspective in P12, L7 as,

‘The capacitance enhancement by confined negative dielectric water could be extended to other systems. For example, H⁺ intercalation, which was not considered in this work, has been reported to exhibit a large capacitance through the protonation of MXene, thereby the influence of water to the H⁺ capacitance would be of particular interest to further increase the capacitance.³³ The influence of nonaqueous electrolyte solvents to the MXene capacitance is also an important issue that needs to be clarified (Supplementary Fig. S20), as it includes more complicated phenomena such as interfacial desolvation, SEI formation, and co-intercalation.^{34,35}’

Reviewer #3 (Remarks to the Author):

- (1) This work focuses on the dielectric constant of electrolyte in the study of supercapacitors and reports the negative dielectric constant of water in a confined slit-shaped space. This study together with the new results is demonstrated the importance for the understanding of supercapacitors and new design of high performance devices particularly for highly volumetric performance ones since the narrowly slit-shaped space is dominant for the reported effect. Considering the new insight of supercapacitors which is supported by some interesting discussion, the referee recommend this submission for publication with some concerns that should be addresses.

Author reply: We appreciate valuable suggestions.

- (2) the authors should give more distinct definitions for negative dielectric constant of water, the physical nature, and the range of the function, etc.

Author reply: We have added the detailed explanation of the negative dielectric constant of water in P10, L2 as,

The electric-double layer formed in MXene system can be modeled as an equivalent circuit of a bilayer capacitor (Fig. 1b). The capacitance of a capacitor sandwiching two series of a contact layer and a water layer can be expressed as $C = \frac{A\epsilon_0}{\lambda}$, where $\lambda = \frac{l^c}{\epsilon_r^c} + \frac{l^h}{\epsilon_r^h}$ with thicknesses (l^c , l^h) and dielectric constants (ϵ_r^c , ϵ_r^h) for a contact layer and a water layer.²⁸ Based on DFT calculations²⁹, in the first term $\frac{l^c}{\epsilon_r^c}$ for the contact layer between the water and an electrode, ϵ_r^c is small to be around 10^0 and l^c is 2–3 Å. In the second term $\frac{l^h}{\epsilon_r^h}$ for the water layer confined in spaces ranging from macroslit ($2b > 50$ nm) to mesoslit ($2 \text{ nm} < 2b < 50$ nm) (Supplementary Table S1), water molecules are weakly bounded to rotate freely and give a large positive ϵ_r^h of approximately 80, where the situation becomes as such $\frac{l^c}{\epsilon_r^c} \gg \frac{l^h}{\epsilon_r^h}$, and hence $\lambda \sim \frac{l^c}{\epsilon_r^c}$.²⁹ Therefore, in macroslit and mesoslit capacitors, the dielectric contact layer (the first term)

dominates the overall capacitance, where the water molecules do not play essential roles. In striking contrast, in a microslit capacitor, the water layer (the second term) largely influences the overall capacitance.’

- (3) the authors is suggested to give a typical example for the design of novel devices highlighting both the gravimetric and volumetric performances, or at least give a more detailed comments on how to use the new finding to design new devices.

Author reply: According to the reveiwer’s comment, we have added comments on the device-design strategy in the conclusion section (P12, L7) as,

‘The capacitance enhancement by confined negative dielectric water could be extended to other systems. For example, H⁺ intercalation, which was not considered in this work, has been reported to exhibit a large capacitance through the protonation of MXene, thereby the influence of water to the H⁺ capacitance would be of particular interest to further increase the capacitance.³³ The influence of nonaqueous electrolyte solvents to the MXene capacitance is also an important issue that needs to be clarified (Supplementary Fig. S20), as it includes more complicated phenomena such as interfacial desolvation, SEI formation, and co-intercalation.^{34,35}’

- (4) please give some discussion what types of confined space is needed for this negative dielectric constant, which must be slit-shaped or not?

Author reply: We suppose that other microporous materials can induce the negative dielectric constant when the porous space is able to confine hydrated Li ions. We have added the comment on this perspective in P12, L16 as,

‘Further studies on similar confinement effects in other microporous materials are expected to stimulate a range of striking discoveries of EDLCs with higher gravimetric and volumetric energy densities.’

(5) since this study add new insights for the design of volumetric devices design, please cite and comment some important references for volumetric supercapacitor study for supercapacitors, like Energy Environ. Sci., 2015, 8, 1390-1403.

Author reply: We have added it as ref. 32.

(6) the references should be formatted (e.g. ref. 7, last and first names should be corrected).

Author reply: We have corrected them.

Reviewers' comments:

Reviewer #1 (Remarks to the Author):

The current version has addressed most previous comments from all reviewers and this reviewer is convinced with most of the responses. For the logical soundness of the 3D-RISM model and its valid corroboration of the experimental observations, the final point below is to be addressed.

1. This question is somewhat related to the Comment 6 of Reviewer #1, as the authors didn't seem to give any explicit answers directly. In the calculation of capacitance (Fig. 1b, Fig. S14, S17 and S18), what is the experimental evidence or theoretical ground for the claimed "intercalation of 0.125 cation per formula unit"? Is it assumed to be true for all Li/Na/Ka/Rb? This seems quite unreasonable unless all ions exhibit the same ΔQ during charge and discharge. Also this value must be rate dependent. The authors should point out or emphasize the appropriate electrochemical test conditions under which the validity of model is guaranteed.

Reviewer #2 (Remarks to the Author):

If the authors thought their finding of "negative dielectric constant of confined water in nanosheets" a general criteria, it should be applicable to another type 2D nanosheets beyond 2D MXene. The investigation on "metallic 1T MoS₂ nanosheet" (Acerce, M., Voiry, D. & Chhowalla, M. Nat. Nanotech. 2015, 10, 313-318 2015) can be conducted to further support the proposed concept. I don't think it is difficult to obtain this material and perform electrochemical tests.

Reviewer #3 (Remarks to the Author):

After a careful examination of the present version, the referee is satisfied with the changes the authors have made.

Reviewer #1 (Remarks to the Author):

- (1) The current version has addressed most previous comments from all reviewers and this reviewer is convinced with most of the responses. For the logical soundness of the 3D-RISM model and its valid corroboration of the experimental observations, the final point below is to be addressed.
1. This question is somewhat related to the Comment 6 of Reviewer #1, as the authors didn't seem to give any explicit answers directly. In the calculation of capacitance (Fig. 1b, Fig. S14, S17 and S18), what is the experimental evidence or theoretical ground for the claimed "intercalation of 0.125 cation per formula unit"? Is it assumed to be true for all Li/Na/Ka/Rb? This seems quite unreasonable unless all ions exhibit the same ΔQ during charge and discharge. Also this value must be rate dependent. The authors should point out or emphasize the appropriate electrochemical test conditions under which the validity of model is guaranteed.

Author reply: '0.125 (= 1/8) cation per formula unit', which allows us to conduct the 3D-RISM calculations using $2 \times 2 \times \sqrt{3}$ rectangular supercell, was based on the experimental values (0.13-0.08 for Li^+ , Na^+ , K^+ , and Rb^+). Perhaps the reviewer #1 is anxious about a small difference from the experimental values. However, as the capacitance is the derivative of ΔQ of the inner potential difference, the use of same ΔQ for all cations does not influence to our discussion/conclusion on the capacitance.

Concerning the rate dependence, we employed the capacitances obtained at the slow scan rate of 0.5 mV/s, excluding the influence of the scan rate. We have added the explanation on the experimental conditions in the main text (P6, L2) as,

'Having determined the surface capacitance of the MXene electrode, we measured the intercalation capacitance using electrolytes consisting of small alkali cations (Li^+ , Na^+ , K^+ , and Rb^+) at the scan rate of 0.5 mV/s.'

Reviewer #2 (Remarks to the Author):

- (1) If the authors thought their finding of "negative dielectric constant of confined water in nanosheets" a general criteria, it should be applicable to another type 2D nanosheets beyond 2D MXene. The investigation on "metallic 1T MoS2 nanosheet" (Acerce, M., Voiry, D. &

Chhowalla, M. *Nat. Nanotech.* 2015, 10, 313-318 2015) can be conducted to further support the proposed concept. I don't think it is difficult to obtain this material and perform electrochemical tests.

Author reply: We appreciate valuable suggestion. Indeed, Geng *et al.* reported that the capacitance of metallic 1T MoS₂ nanosheet with an aqueous Li⁺ electrolyte is larger than that with an aqueous Na⁺ electrolyte (X. Geng *et al.*, *Nano Lett.*, (2017) 17, 1825). Our finding reasonably explains this capacitance enhancement, proving its generality. We have added the explanation in the main text (P12, L6) with this reference as,

‘Furthermore, Geng *et al.* reported that the capacitance of metallic 1T MoS₂ with an aqueous Li⁺ electrolyte is larger than that with an aqueous Na⁺ electrolyte.³²,

REVIEWERS' COMMENTS:

Reviewer #1 (Remarks to the Author):

The authors have addressed the reviewers' comments further. It can be accepted at this stage.

Reviewer #2 (Remarks to the Author):

The authors answered my questions. The manuscript can be accepted after addressing other reviewers' comments.